# Optimization of Autohydrolysis of Olive Pomaces to Obtain Bioactive Oligosaccharides: The Effect of Cultivar and Fruit Ripening

**Laura Freitas [1], Rita Simões [2], Isabel Miranda [2], Fátima Peres [1,3] and Suzana Ferreira-Dias [1,*]**

[1] Associated Laboratory TERRA, LEAF—Linking Landscape, Environment, Agriculture and Food—Research Center, Instituto Superior de Agronomia, Universidade de Lisboa, 1349-017 Lisbon, Portugal; freitas_laura97@live.com.pt (L.F.); fperes@ipcb.pt (F.P.)

[2] Associated Laboratory TERRA, Centro de Estudos Florestais, Instituto Superior de Agronomia, Universidade de Lisboa, 1349-017 Lisbon, Portugal; ritafabiana@isa.ulisboa.pt (R.S.); imiranda@isa.ulisboa.pt (I.M.)

[3] Instituto Politécnico de Castelo Branco, Escola Superior Agrária, 6001-909 Castelo Branco, Portugal

* Correspondence: suzanafdias@mail.telepac.pt

**Abstract:** The valorisation of agro-industrial residues presents a challenge in obtaining economically sustainable and environmentally friendly industrial processes. Olive pomace is a by-product generated in large quantities, from olive oil extraction. This residue mostly consists of lignocellulosic materials. The aim of this study was to evaluate the potential use of extracted olive pomaces (EOP) obtained from olives with different ripening indexes (RI) and from different cultivars (Cobrançosa; RI = 2.5; 3.3 and 4.7; and Galega Vulgar; RI = 1.8; 2.9 and 4.8), to produce bioactive oligosaccharides from hemicelluloses by autohydrolysis. The hydrothermal treatment conditions were optimized by Response Surface Methodology, following a central composite rotatable design (CCRD), as a function of temperature (T: 142–198 °C) and time (*t*: 48–132 min), corresponding to severity factor (SF) values from 3.2 to 4.9. For all pomace samples, soluble sugar production was described by concave surfaces as a function of temperature and time. Autohydrolysis with SF equal or higher than 4.0 produced higher sugar yields, with maximum values around 180 g glucose equivalent/kg EOP for SF of 4.7 (190 °C/120 min) or 4.9 (198 °C/90 min). These values were similar for both cultivars and were not dependent on the ripening stage of the olives. Maximum oligosaccharide (OS) yields of 98% were obtained by autohydrolysis with SF of 4.0. The increase in SF to 4.9 resulted in a decrease in OS yield to 86–92%, due to the release of monomeric sugars. The monosaccharides were mostly xylose (55.8–67.7% in Galega; 50.4–69.0% in Cobrançosa liquid phases), and glucose, galactose, arabinose and rhamnose, in smaller quantities. Therefore, the production of bioactive xylo-oligosaccharides (XOS) from olive pomaces mainly depends on the hydrothermal conditions used.

**Keywords:** autohydrolysis; oligosaccharides; olive pomace; olive ripening; optimization; prebiotic sugars; response surface methodology; xylo-oligosaccharides

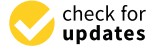

## 1. Introduction

In the Mediterranean diet, the main source of lipids is provided by the consumption of virgin olive oil which has scientifically recognized health benefits in addition to its unique flavour properties [1]. Due to these properties, olive oil production is no longer restricted to the Mediterranean Basin but has been expanding globally to meet consumer demands. In fact, its production has tripled in the last 60 years and, according to the International Olive Council, a production of 3,098,500 *t* of olive oil is estimated for the 2021/2022 crop year [2]. The EU (Spain, Italy, Greece, and Portugal, by order of production) accounts for 64% of global production, where Spain alone is responsible for 66% of EU production.

Olive oil is only extracted by mechanical processes. In brief, olives after milling and malaxation, are submitted to pressing (in traditional olive mills) or centrifugation, in two-

or three-phase decanters, to extract the virgin olive oil from olive pulp. Irrespective of the extraction mill system used (pressing or centrifugation), this industry generates large quantities of solid wastes, i.e., olive pomaces which contain 3–4.5% residual oil (wet basis), olive pulp, stones and water. Pomaces from press or three-phase decanters have a lower water content, namely around 20–25% and 40 to 50%, respectively, than those obtained in two-phase systems (up to 70%) [3–5]. In fact, the pomaces from two-phase decanter systems consist of a semi-solid paste containing the solid fraction and the water (processing water and vegetation water) that is a separated stream in traditional and three-phase decanters olive mills (olive mill wastewater). Multi-phase decanters (DMF) are an innovation in the two-phase extraction system, that work without adding process water and have the advantage of recovering a dried pomace similar to that coming from the three-phase decanters (45–55% moisture content). This pomace is called paté which consists of olive pulp and vegetation water, without traces of stone. This represents another way to obtain a product with more added value.

It is estimated that over $12 \times 10^6$ tonnes of wet pomace were discarded worldwide in 2020/2021, corresponding to over $5.5 \times 10^6$ tonnes of dry lignocellulosic materials. These values were obtained considering an olive oil extraction yield of 20% (m/m oil/fresh olives), and a pomace with 55% moisture content and 3% oil content.

Olive pomace is a lignocellulosic containing material with potential for valorisation, following a biorefinery concept, as a source of oil, lignocellulosic material, renewable energy, nutrients, and bioactive compounds for innovative functional foods [6,7].

Currently, the residual oil is solvent-extracted after pomace drying. After refining, it is used as an edible oil (olive pomace oil). Prior to olive pomace oil extraction, most of the olive stones are removed to decrease its initial concentration of around 45% to less than 15% [8]. The separated fraction of olive stones [8] is used as an energy source [9].

Recently, the use of pomaces to produce bio-oil by fast pyrolysis has been investigated [10]. Olive pomaces from two-phase systems result in a high content of potassium and have been tested as fertilizers [9]. However, there is the risk of some plant toxicity [9]. Moreover, olive pomace is a source of bioactive phenolic compounds with recognized antioxidant and antimicrobial properties [7,11–15].

Extracted olive pomace (EOP) and stones mainly consist of cellulose, hemicelluloses and lignin. The extracted olive pomace typically has cellulose contents in the range of 13.8–30.0%, 18.5–32.2% hemicelluloses and 30.0–41.6% lignin [16–19]. The olive stones contain cellulose content in the range of 20.1–40.4%, 18.5–32.5% of hemicelluloses, and 25.3–48.4% of acid insoluble lignin [19–23]. Cellulose is used as source of glucose for bioethanol production or other uses [17,19,22,24,25]. The hemicelluloses in olive pomace and olive stones are rich in xylans, and xylose, which account for approximately 52% of the total monomeric units in olive pomace and 20.6–44.4% of total monomeric units in olive stones [19,26].

Xylo-oligosaccharides have been recognized as bioactive compounds with prebiotic activity [6]. Xylose monomers from olive stones are used as raw material for (i) xylitol production, a natural sweetener for food products, by fermentation with *Pachysolen tannophilus* [26] and (ii) to produce furfural, in a microwave reactor, using $FeCl_3$ and sulfuric acid as catalysts [27,28].

The lignin fraction of pomace and stones, which consists of aromatic structures of monolignols, is recalcitrant and presents a structural heterogeneity. Its extraction and purification is needed for waste biomass conversion into added-value chemicals [6].

However, the use of these different components of lignocellulosic materials (hemicelluloses, cellulose, and lignin) needs efficient pre-treatments for biomass fractionation. Therefore, extracted olive pomaces (EOP) or stones must first be submitted to a thermal treatment to hydrolyse the hemicelluloses into oligosaccharides or sugar monomers solubilized in a liquid stream (liquor) and to obtain a solid fraction mainly formed by cellulose and lignin. In most of the cases, the first step consists of dilute acid hydrolysis [17,26–28] in the presence of sulfuric acid, which is a pollutant process generating acidic effluents.

As an alternative, hydrothermal pre-treatments (e.g., autohydrolysis and steam explosion) are eco-friendly processes to promote hydrolysis of hemicelluloses and to enable the subsequent deconstruction of cellulose into glucose monomers by enzyme-catalysed hydrolysis (saccharification).

Autohydrolysis has been carried out to produce oligosaccharides and sugars from EOP [19,29,30], or olive stones [19,22]. The extent of hemicellulose hydrolysis depends on the treatment conditions (temperature and time). However, in these studies, industrial pomaces without any information about the olive cultivar or ripening stage of the olives were disclosed. Moreover, limited information about the combined effect of autohydrolysis temperature and time on autohydrolysis of hemicelluloses is given.

Therefore, the aim of this study was to evaluate the effect of autohydrolysis conditions (time and temperature) on hemicellulose hydrolysis from olive pomaces from two Portuguese cultivars ('Galega Vulgar' and 'Cobrançosa') obtained from olive oil extraction during olive ripening. The optimization of autohydrolysis conditions, aimed at the production of soluble sugars, namely bioactive oligosaccharides, and monomers, was carried out using response surface methodology.

## 2. Results and Discussion

### 2.1. Chemical Characterization of the Olives

The initial moisture and oil content of the olives were assayed immediately after harvest. The results are listed in Table 1. The moisture content in Galega Vulgar cultivar increased during ripening, while the moisture content of Cobrançosa remained constant during the harvest period, with a decrease at the highest ripening index (RI). Concerning oil content, the values ranged from 37.7 to 45.7% (d.b.) in Galega olives, and from 35.23 to 36.29% (d.b.) in Cobrançosa olives. For both cultivars, the oil content at the highest ripening stage (RI = 4.8 for Galega and 4.7 for Cobrançosa olives) was significantly higher than the values at the initial ripening stages. This increase in oil content during ripening, as well as the range of oil content for both cultivars, are reported by Beltrán et al. (2004) and Peres et al. (2016) [31,32].

**Table 1.** Moisture and oil content (d.b.) of Galega Vulgar or Cobrançosa olive fruits prior to olive oil extraction. The results are the average of four values ± standard deviation. Means, in the same column, followed by the same letters are not significantly different ($p > 0.05$; post-hoc Scheffé test).

| Cultivar | CODE | Ripening Index | Moisture (%) | Oil (%, d.b.) |
|---|---|---|---|---|
| Galega Vulgar | GAL-1 | 1.9 | 54.26 ± 0.197 [a] | 37.7 ± 0.203 [a] |
| | GAL-2 | 2.9 | 54.80 ± 0.135 [a] | 38.9 ± 0.234 [a] |
| | GAL-3 | 4.8 | 60.01 ± 0.068 [b] | 45.7 ± 0.333 [b] |
| Cobrançosa | COB-1 | 2.5 | 55.71 ± 0.135 [c] | 35.46 ± 0.362 [c] |
| | COB-2 | 3.3 | 55.36 ± 0.145 [c] | 35.23 ± 0.352 [c] |
| | COB-3 | 4.7 | 54.54 ± 0.133 [d] | 36.29 ± 0.258 [d] |

### 2.2. Chemical Characterization of the Olive Pomaces

After olive oil extraction using an Abencor extractor system, the moisture content of the olive pomace was assayed. Galega pomace showed a decrease in the water content from 42.6 to 38.6% during ripening (Table 2). The moisture content of Cobrançosa cultivar pomace was lower than the Galega pomace, as observed in the fruits (Table 1). Again, for both cultivars, only the water content of the pomaces obtained from fruits with the highest RI was significantly higher than the values for the other pomaces.

**Table 2.** Chemical composition (% d.w.) of pomaces from Galega (GAL) and Cobrançosa (COB) olives with different ripening indexes (RI). Means, in the same row, followed by the same letters are not significantly different ($p > 0.05$; post-hoc Scheffé test).

| Components | Galega Vulgar | | | Cobrançosa | | |
|---|---|---|---|---|---|---|
| | GAL-1 (RI: 1.9) | GAL-2 (RI: 2.9) | GAL-3 (RI: 4.8) | COB 1 (RI: 2.5) | COB 2 (RI: 3.3) | COB 3 (RI:4.7) |
| Water | 42.6 | 40.6 | 38.6 | 39.3 | 36.7 | 36.5 |
| Ash | 4.10 ± 0.023 [a] | 3.90 ± 0.022 [a] | 3.75 ± 0.041 [b] | 3.62 ± 0.137 [a] | 3.35 ± 0.003 [a] | 3.33 ± 0.086 [a] |
| Total extractives: | 69.64 ± 2.17 [a] | 70.61 ± 1.03 [a] | 69.51 ± 1.61 [a] | 57.72 ± 2.33 [c] | 57.88 ± 0.55 [c] | 64.14 ± 2.99 [c] |
| • *n*-hexane | 25.19 ± 0.58 [a] | 26.85 ± 0.41 [b] | 40.56 ± 0.32 [c] | 24.18 ± 0.41 [a] | 22.13 ± 2.36 [a] | 31.16 ± 0.64 [b] |
| • Ethanol | 25.52 ± 0.96 [a] | 31.81 ± 1.71 [b] | 13.63 ± 0.99 [c] | 21.44 ± 2.14 [a] | 21.42 ± 2.10 [a] | 22.05 ± 2.43 [a] |
| • Water | 18.46 ± 3.19 [a] | 11.95 ± 1.27 [a] | 15.33 ± 1.96 [a] | 12.10 ± 0.21 [a] | 14.59 ± 0.52 [b] | 10.92 ± 0.14 [c] |
| Cutin | 2.72 ± 0.20 [a] | 1.70 ± 0.18 [a] | 3.55 ± 0.50 [a] | 2.48 ± 0.08 [b] | 2.22 ± 0.04 [b] | 1.80 ± 0.15 [c] |
| Total lignin: | 10.91 ± 0.30 [a] | 10.37 ± 0.29 [a] | 10.49 ± 0.57 [a] | 15.52 ± 1.12 [b] | 15.29 ± 0.71 [b] | 12.96 ± 0.94 [c] |
| • Klason lignin | 10.69 ± 0.31 [a] | 10.19 ± 0.30 [a] | 10.40 ± 0.56 [a] | 15.38 ± 1.13 [a] | 15.17 ± 0.07 [a] | 12.83 ± 0.89 [b] |
| • Soluble lignin | 0.22 ± 0.02 [a] | 0.17 ± 0.01 [b] | 0.09 ± 0.01 [c] | 0.14 ± 0.004 [a] | 0.12 ± 0.003 [b] | 0.14 ± 0.01 [a] |
| Polysaccharides | 14.20 ± 164 [a] | 13.92 ± 0.97 [a] | 13.28 ± 0.88 [a] | 20.56 ± 1.51 [b] | 21.00 ± 0.58 [b] | 17.77 ± 206 [b] |

The chemical composition of pomaces from Galega and Cobrançosa cultivars at different RI is presented in Table 2. The olive pomaces, obtained from the Abencor extractor system, are characterized by a high residual oil content [33]. The residual oil fraction, extracted with *n*-hexane, represented 25–26% (d.w.) of Galega pomaces GAL-1 and GAL-2 but greatly increased in GAL-3 (40.56%). For Cobrançosa pomaces, the value also significantly increased in sample COB-3. Therefore, this residual oil fraction in both pomaces increased with ripening. It seems that it is easier to extract olive oil from ripe fruits. However, the Abencor system yield is not comparable to that of industrial mills. Galega olives were already referred as an example of difficult pastes in malaxation. Therefore, different strategies must be applied in industrial mills to have low residual oil in these pomaces [33].

The contents of extractives soluble in ethanol and water, which include phenolics and polyphenolics [15,30], corresponded to a high proportion of the total dry mass, in both Galega and Cobrançosa pomaces. For Galega pomaces, the total of extractives in ethanol and water was approximately 44% in the samples with RI of 1.9 and 2.9. A significant decrease of about 34% was observed in the extractives for the pomace with the highest RI (GAL-3: 29.96%, d.w.). With respect to Cobrançosa pomace, no significant differences were found in the content of extractives soluble in ethanol and water during ripening, presenting an average value of 34.18% (46.07% is considered in an oil-free pomace). The significant differences found in the concentrations of polar extractives in Galega and Cobrançosa pomaces might be related to olive cultivar and to the effect of fruit ripening on the formation of phenolic compounds. These results are in accordance with the results found in extracted olive pomaces by Gómez-Cruz et al. (2020) [15] who reported 46.8% acetone–water extractives, and Manzanares et al. (2020) [30] who reported 42.0% ethanol–water extractives.

The mean ash content was 3.9 and 3.4% in Galega and Cobrançosa pomaces, respectively (Table 2). These values are similar to the value reported by Fernandes et al. (2016) [17] of 4.4%, but lower than the values of 6.0 and 7.3%, reported by Ruggeri et al. (2015) [16], and Miranda et al. (2019) [19], respectively. Galega and Cobrançosa pomaces contained approximately 2.6 and 2.1% cutin, respectively, which comes from the olive skin.

The structural components of Galega and Cobrançosa pomaces contained approximately 10 and 15% lignin, respectively, and 13.8 and 19.8% structural carbohydrates (corresponding to an extractive and cutin-free basis between 32.4 and 35% lignin and

44.5 and 49.5% carbohydrate content), respectively. Concerning the ripening effect on lignin and polysaccharide contents, no significant variation was observed for Galega pomace. In Cobrançosa pomace, a significant decrease in both structural components was observed for the sample with the highest RI (COB-3). The following values for the composition of extracted olive pomace samples from non-specified cultivars, obtained from industrial olive mills, and partly destoned before pomace oil solvent-extraction, were reported: Miranda et al. (2019) [19] reported 31.2% lignin and 36.5% polysaccharides (3.8% glucan and 22.7% hemicelluloses rich in xylans), Fernandes et al. (2016) [17] reported 33.9% Klason lignin, 23.3% hemicelluloses and 22.9% glucan, Leite et al. (2016) [18] reported 41.6% lignin and 35.3% polysaccharides (24.1% hemicelluloses and 11.2% cellulose), and Ruggeri et al. (2015) [16] reported 37% lignin, and 49.5% polysaccharides.

### 2.3. Autohydrolysis Experiments and Modelling

The autohydrolysis experiments were carried out for the six extracted olive pomace (EOP) samples, under the conditions dictated by the experimental matrix (Table 3). The quantities of soluble sugars, expressed as g glucose equivalent per kg of dry extracted pomace, obtained for each CCRD experiment and EOP, are shown in Table 3, as well as the corresponding severity factors (SF) of each autohydrolysis conditions.

**Table 3.** Sugar production (g glucose equivalent/kg dry extracted pomace) by autohydrolysis of extracted olive pomaces from Galega Vulgar (GAL) and Cobrançosa (COB) cultivars, obtained from fruits at different ripening stages (see Table 2), carried out under different experimental conditions and severity factors (SF).

| Run $n °$ | Temperature (°C) | Time (min) | GAL-1 (g Glu/kg) | GAL-2 (g Glu/kg) | GAL-3 (g Glu/kg) | COB-1 (g Glu/kg) | COB-2 (g Glu/kg) | COB-3 (g Glu/kg) | SF (Log $R_0$) |
|---|---|---|---|---|---|---|---|---|---|
| 1 | 150 | 60 | 22.0 | 31.2 | 28.6 | 58.1 | 36.0 | 52.1 | 3.3 |
| 2 | 150 | 120 | 107.2 | 107.6 | 56.9 | 105.2 | 88.8 | 98.1 | 3.6 |
| 3 | 190 | 60 | 177.5 | 148.9 | 165.0 | 151.6 | 153.3 | 168.6 | 4.4 |
| 4 | 190 | 120 | 107.8 | 171.9 | 180.3 | 176.4 | 168.2 | 183.5 | 4.7 |
| 5 | 142 | 90 | 27.0 | 29.6 | 26.9 | 21.2 | 15.2 | 41.5 | 3.2 |
| 6 | 198 | 90 | 127.6 | 177.8 | 114.3 | 178.3 | 141.1 | 161.7 | 4.9 |
| 7 | 170 | 48 | 142.0 | 168.5 | 130.3 | 72.4 | 112.8 | 140.4 | 3.7 |
| 8 | 170 | 132 | 163.4 | 107.8 | 149.9 | 175.6 | 179.1 | 148.9 | 4.2 |
| 9 | 170 | 90 | 150.6 | 165.0 | 149.1 | 167.5 | 151.3 | 153.4 | 4.0 |
| 10 | 170 | 90 | 168.9 | 177.5 | 160.2 | 145.1 | 142.8 | 164.7 | 4.0 |
| 11 | 170 | 90 | 169.9 | 158.2 | 165.7 | 146.2 | 139.6 | 169.8 | 4.0 |
| 12 | 170 | 90 | 165.6 | 166.7 | 157.3 | 152.6 | 170.0 | 149.3 | 4.0 |

The highest sugar yields were obtained when autohydrolysis was carried out under reaction conditions with high severity factors (SF ≥ 4.0). In fact, maximum sugar yields were: 177.5 g/kg dry pomace for GAL-1 at 190 °C/60 min (SF = 4.4); 177.8 g/kg dry pomace for GAL-2, at 198 °C/90 min (SF = 4.9); 180.3 g/kg dry pomace for GAL-3, at 190 °C/120 min (SF = 4.7); 178.3 g/kg dry pomace for COB-1, at 198 °C/90 min (SF = 4.9); 179.1 g/kg dry pomace for COB-2, at 170 °C/132 min (SF = 4.2); and 183.5 g/kg dry pomace for COB-3, at 190 °C/120 min (SF = 4.7). This shows that applying increased temperature, and prolonged times, will result in the decomposition of polysaccharides into OS and monomers. The concentration of soluble sugars obtained by autohydrolysis of pomaces of both cultivars and from fruits at different ripening stages were similar.

The results presented in Table 3 were used to calculate the linear and quadratic effects of temperature (T) and time (*t*), as well as the interaction $t \times T$, on soluble sugar production by autohydrolysis, for all pomace samples. The factors with significant effects ($p \leq 0.05$) or those that, even with a *p*-value higher than 0.05, when removed, resulted in a lack of fit of the model, were considered in the polynomial second-order models fitted to the experimental data points [34,35]. The model equations and respective determination coefficients ($R^2$) and adjusted $R^2$ ($R^2_{adj}$) are shown in Table 4. For all pomaces of both cultivars, obtained from fruits with different ripening stages, sugar production by autohydrolysis could be

described by second-order polynomial models, as a function of reaction temperature and operation time (linear and/or quadratic effects).

**Table 4.** Polynomial equations of the models fitted to experimental results of autohydrolysis of EOP as a function of temperature (T, °C) and time (*t*, min) and respective $R^2$ and $R^2_{adj}$. GAL-1, GAL-2 and GAL-3 correspond to Galega pomaces obtained from olives at different ripening stages. COB-1, COB-2 and COB-3 correspond to Cobrançosa pomaces obtained from olives at different ripening stages (see Table 2); glucose equivalent concentration, [Glu], is expressed in g/kg EOP and decoded values of the factors are considered in the equations.

| EOP | Model Equation | $R^2$ | $R^2$adj |
|---|---|---|---|
| GAL-1 | $\lfloor Glu \rfloor = -4791.72 + 49.35\ T - 0.12\ T^2 + 13.27\ t - 0.01\ t^2 - 0.06\ Txt$ | 0.992 | 0.985 |
| GAL-2 | $\lfloor Glu \rfloor = -2496.5 + 28.78\ T - 0.08\ T^2 - 0.01\ t^2 + 0.01\ Txt$ | 0.799 | 0.682 |
| GAL-3 | $\lfloor Glu \rfloor = -3514.48 + 39.6\ T - 0.11\ T^2 + 1.95\ t - 0.01\ t^2$ | 0.925 | 0.883 |
| COB-1 | $\lfloor Glu \rfloor = -2093.32 + 21.89\ T - 0.06\ T^2 + 3.02\ t - 0.01\ t^2$ | 0.944 | 0.913 |
| COB-2 | $\lfloor Glu \rfloor = -3012.61 + 32.66\ T - 0.08\ T^2 + 3.36\ t - 0.02\ Txt$ | 0.993 | 0.989 |
| COB-3 | $\lfloor Glu \rfloor = -2723.66 + 29.25\ T - 0.08\ T^2 + 4.17\ t - 0.01\ t^2 - 0.01\ Txt$ | 0.977 | 0.958 |

Concerning Galega pomaces, sugar production showed to depend on all the factors considered (linear and quadratic effects) as well as on the interaction $t \times T$, for GAL-1 and GAL-2. For olive pomaces from olives with higher RI (GAL-3), the interaction effect showed not to be significant. GAL-1 and GAL-3 models exhibited a very high fit to the experimental results with determination coefficients higher than 0.90 [34]. This means that more than 90% of the variability of the data was explained by the model. The polynomial model for GAL-2 showed a high fit ($R^2 > 0.75$) to the experimental results [34].

Moreover, for Cobrançosa pomaces, both temperature and time showed significant effects (linear and quadratic) on sugar production by autohydrolysis (Table 4). For COB-1, the interaction effect between t and T was not significant, and therefore, was removed in the fitted model equation. For all Cobrançosa models, both $R^2$ and $R^2_{adj}$ had values higher than 0.90, indicating a very good fit to the experimental results.

A graphical representation of the second-order polynomial equation models, fitted to the experimental results, was performed by response surface methodology (RSM). The negative coefficients of the quadratic terms of these polynomial equations indicate that sugar production can be described by convex surfaces as a function of autohydrolysis temperature and/or reaction time, as observed in Figure 1, for Galega pomaces, and Figure 2, for Cobrançosa pomaces.

Response surfaces presented similar shapes and values for both cultivars. In all situations, the effect of temperature was more important than the effect of time of autohydrolysis in sugar production. Moreover, the shape of the response surfaces fitted to the experimental data points shows that the curvature is more pronounced in relation to the temperature axes than to the time axes. The ripening stage of the fruits seemed not to affect the soluble sugar quantity, which are mainly obtained by autohydrolysis of hemicelluloses. This suggests that lignocellulosic materials are formed in the olive fruit before ripening starts. In fact, Table 2 shows that, for each cultivar, the content of polysaccharides in the pomaces did not significantly vary during ripening. However, a reduction in soluble sugars was reported in water-stressed samples, suggesting an acceleration of the ripening process [36].

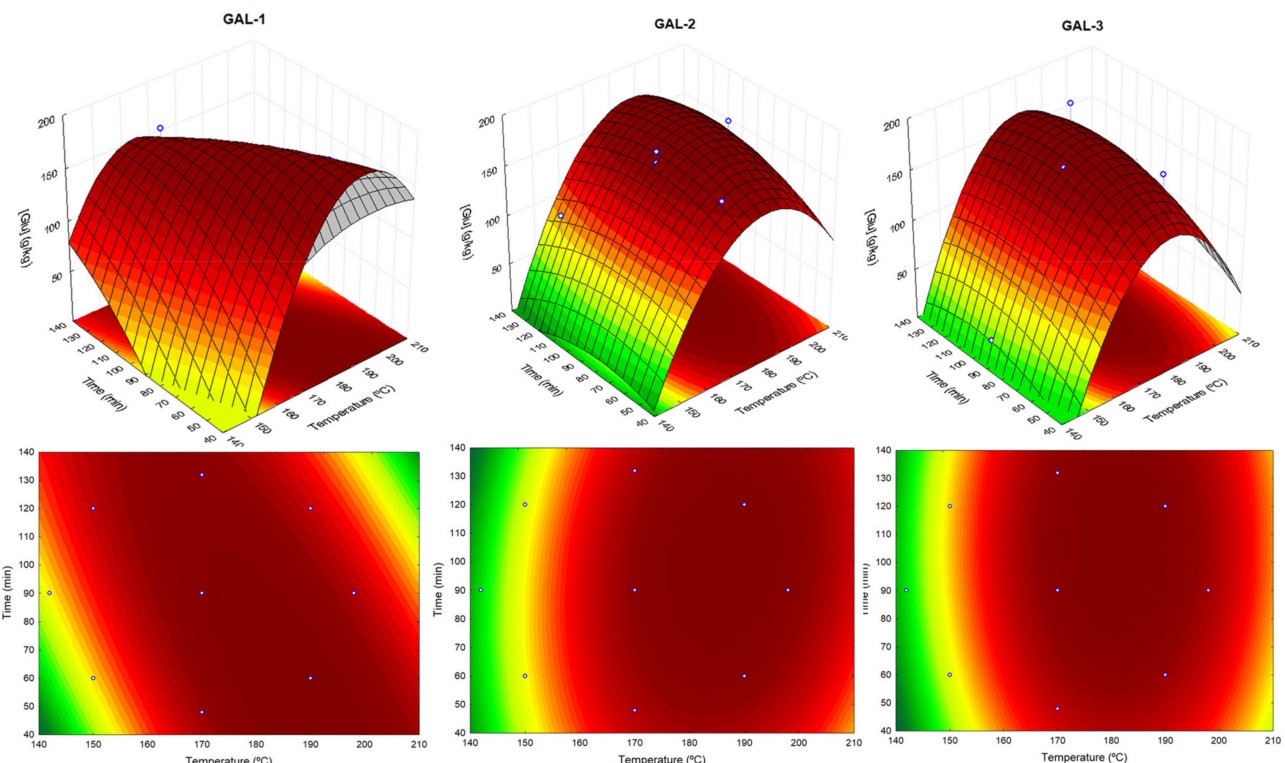

**Figure 1.** Three-dimensional response surface plots, and respective contour plots, of soluble sugar production (g glucose equivalent per kg of extracted olive pomace), from Galega pomaces, as a function of autohydrolysis conditions (temperature and time). GAL-1, GAL-2 and GAL-3 correspond to Galega pomaces obtained from olives at different ripening stages (see Table 2).

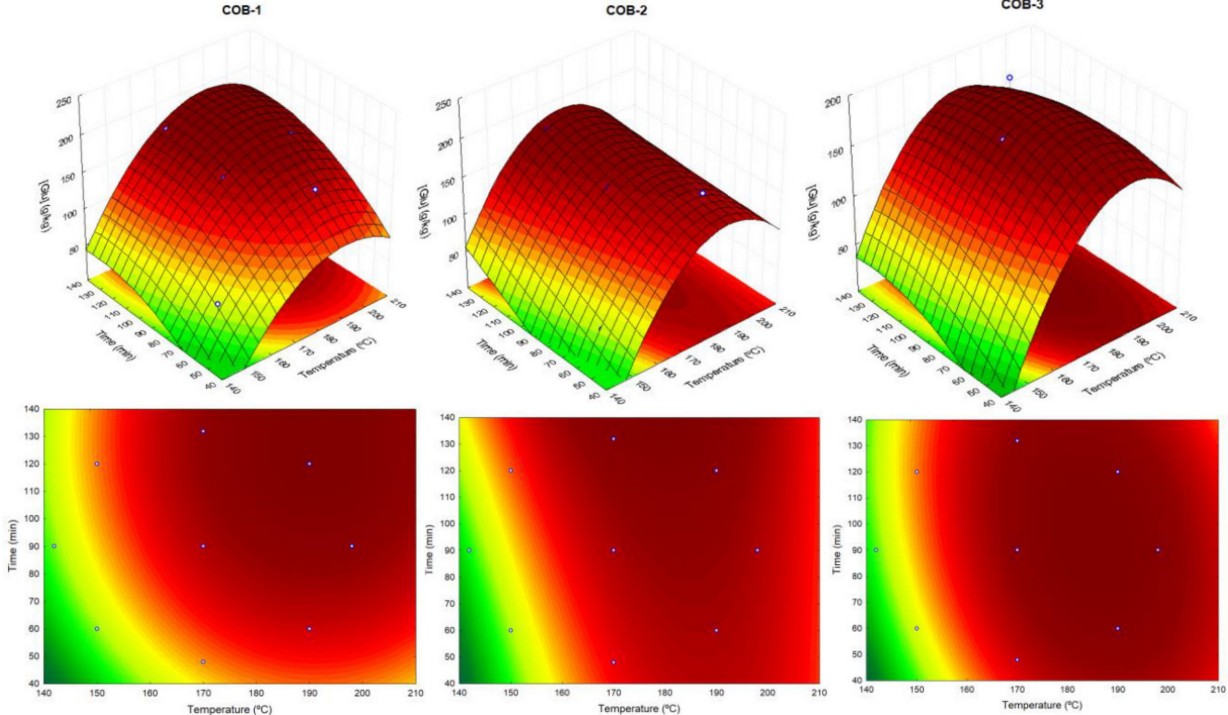

**Figure 2.** Three-dimensional response surface plots, and respective contour plots, of soluble sugar production (g glucose equivalent per kg of extracted olive pomace), from Cobrançosa pomaces, as a function of autohydrolysis conditions (temperature and time). COB-1, COB-2 and COB-3 correspond to Cobrançosa pomaces obtained from olives at different ripening stages (see Table 2).

For each second-order polynomial model equation in Table 4, the optimal conditions of time and temperature to maximize sugar production can be determined by partial differentiation of the polynomial equation and equating the derivatives to zero [34,35]. The mathematical solution found for each system of equations is the stationary point. However, from a technological point of view, it is more important to define the variation range for temperature and time that will lead to high sugar production, i.e., the best operation region, than to have a single point. Analysing the response surfaces and contour plots for the different olive pomaces submitted to autohydrolysis, the best operational regions, in terms of temperature and time, were identified (Figures 1 and 2). Therefore, the best temperature, to attain higher sugar yields, appears to be around 175–200 °C for all the pomaces. For Galega pomace, reaction time must be between 70 and 120 min. For Cobrançosa pomaces, high sugar yields were obtained for hydrolysis durations from 110 to 130 min (COB-1 and COB-2) and from 70 to 130 min, for COB-3. The identified operation conditions correspond to high severity treatments, i.e., SF ≥ 4.0, confirming the results in Table 3.

Model validation was performed for Galega (GAL-2) and Cobrançosa (COB-2) pomace autohydrolysis at 180 °C for 120 min, corresponding to a severity factor of 4.4. These samples were chosen because they correspond to the optimal ripening index for Galega and Cobrançosa olive harvest to obtain virgin olive oils rich in bioactive compounds [32,37]. The results obtained were 170.8 ± 17.61 and 229.0 ± 30.62 g Glu/kg, for Galega and Cobrançosa pomaces, respectively. The predicted values by the models are 163.9 and 245.4 g Glu/kg pomace, respectively. The obtained results confirm the good fit of the models to the experimental results.

No values were found in the literature, on the optimization of autohydrolysis of monocultivar olive pomaces, using response surface methodology, as a function of time and temperature conditions. However, several studies have been conducted on thermal treatments of olive pomaces, most of them obtained from industrial pomace oil extraction plants, namely non-isothermal autohydrolysis of EOP (150–230 °C) [29], dilute acid hydrolysis using 3.5% sulfuric acid at 130 °C for 130 min [17], and autohydrolysis at 130 °C/130 min [19]. Fernandes et al. (2012) [29] concluded that the temperature of the autohydrolysis of EOP under non-isothermal conditions should be higher than 170 °C to be effective in the breakdown of hemicelluloses. When EOP and stones were submitted to autohydrolysis under decreased temperatures, i.e., 130 °C/130 min, corresponding to a severity factor of 3.0, only 65% of hemicelluloses in EOP and 75% in olive stones were hydrolysed into oligosaccharides, which were mainly xylo-oligosaccharides, while neither cellulose nor lignin underwent hydrolysis [19]. Cuevas et al. (2015) [22] tested different autohydrolysis conditions on olive stones, with temperatures varying from 150 to 225 °C and operation times up to 10 min. An autohydrolysis of a severity factor equal to 3.7 resulted in the complete solubilization of hemicelluloses, while the cellulose and lignin fractions were preserved; the highest oligosaccharide yield was observed for a treatment with a severity factor of 3.59 (190 °C/5 min) [22]. These studies were aimed to release hemicelluloses and to obtain a solid fraction, rich in cellulose and lignin, which was used for saccharification [17,19,22,29]. Moreover, when industrial extracted olive pomace was used, partial removal of stones was performed before pomace drying and solvent extraction of the oil. Olive stones are a recalcitrant material due to their higher lignin content (42.1%, d.w.) and approximately 2.5 times less extractives than that of extracted pomaces, partially destoned before processing (31.2% lignin, d.w.). Therefore, more intense autohydrolysis conditions must be applied for stones [19]. In our case, where destoned pomaces were not used, higher amounts of olive stones were present. However, our results are in accordance with those previously reported for autohydrolysis of pomaces and stones [19].

### 2.4. Monomeric and Oligosaccharides Quantification

Table 5 presents the identification and quantification of neutral monomeric sugars, mainly derived from hemicelluloses, in Cobrançosa and Galega liquid phases obtained by autohydrolysis of pomaces at 170 °C/90 min (central point; SF = 4.0), 180 °C/120 min

(model validation conditions for GAL-2 and COB-2 EOP; SF = 4.4) and 198 °C/30 min (star point with the highest temperature; SF = 4.9). These results, together with the quantification of total soluble sugars, were used to estimate oligosaccharide (OS) yields (cf. 3.4.).

**Table 5.** Sugar composition (monomeric sugars and oligosaccharides, OS) of the liquid phase resulting from autohydrolysis treatments of EOP under different conditions of temperature (T) and time (*t*).

| EOP | Autohydrolysis | Total Sugars Released | Monomeric Sugars (% Total Sugars) | | | | OS | OS Yield |
|---|---|---|---|---|---|---|---|---|
| | (T/*t*; °C/min) | (g/kg EOP, d.w.) | Rhamnose | Arabinose | Glucose | Xylose | (g/kg EOP, d.w.) | (%) |
| GAL-1 | 170/90 | 150.6 | 2.85 | 5.06 | 24.38 | 55.28 | 147.90 | 98.21 |
| GAL-1 | 198/90 | 127.6 | 5.01 | 23.96 | 2.55 | 63.90 | 110.23 | 86.38 |
| GAL-2 | 170/90 | 165.9 | 3.22 | 4.98 | 20.95 | 61.04 | 162.16 | 97.74 |
| GAL-2 | 180/120 | 170.8 | 0.00 | 12.67 | 12.02 | 65.36 | 136.23 | 79.86 |
| GAL-2 | 198/90 | 177.8 | 5.03 | 20.06 | 2.93 | 67.66 | 163.78 | 92.12 |
| GAL-3 | 170/90 | 149.1 | 3.44 | 2.94 | 26.94 | 54.93 | 146.61 | 98.33 |
| GAL-3 | 198/90 | 114.3 | 7.60 | 24.22 | 5.52 | 57.73 | 100.71 | 88.11 |
| COB-1 | 170/90 | 167.5 | 2.48 | 5.05 | 28.80 | 50.42 | 163.86 | 97.83 |
| COB-1 | 198/90 | 178.3 | 5.90 | 28.82 | 3.01 | 57.68 | 156.71 | 87.89 |
| COB-2 | 170/90 | 151.3 | 2.34 | 4.16 | 25.87 | 56.22 | 148.40 | 98.09 |
| COB-2 | 180/120 | 229.0 | 0.00 | 15.17 | 8.38 | 68.99 | 197.32 | 86.16 |
| COB-2 | 198/90 | 141.1 | 5.24 | 23.65 | 2.81 | 64.13 | 120.97 | 85.73 |
| COB-3 | 170/90 | 153.4 | 2.50 | 4.02 | 32.92 | 51.23 | 149.53 | 97.48 |
| COB-3 | 198/90 | 161.7 | 6.60 | 23.97 | 4.14 | 60.78 | 143.70 | 88.87 |

The liquid phases resulting from hydrothermal treatments contain a mixture of oligomeric compounds and monosaccharides. Sugars are predominantly in oligomeric form varying between 80 and 98% of the total sugars, in Galega liquid phases, and from 86 to 98%, in Cobrançosa liquid phases, depending on the severity conditions of the treatment.

The concentration of monomeric compounds increased from 2.41 to 34.6 g/kg EOP (data not shown). The monomeric fraction mainly contained xylose (55.3–67.7% in Galega and 50.4–69.0% in Cobrançosa liquid phases), and decreased quantities of glucose, galactose, arabinose and rhamnose (Table 5). For each pomace sample, the increase in the severity factor of the autohydrolysis (SF from 4.0 to 4.9) resulted in an increase in xylose. The hydrolysis leads to a slight decrease in OS formation. In fact, the OS yields ranged from 97.7–98.3%, in Galega, and 97.5–98.1% in Cobrançosa liquid phases obtained from autohydrolysis with a severity factor of 4.0 (170 °C/90 min) but decreased to between 86.4 to 92.1% and 85.7 to 88.9%, when Galega and Cobrançosa pomaces were auto-hydrolysed at 198 °C/90 min (SF = 4.9).

The concentration of monomeric xylose reached the highest value at SF = 4.9 (198 °C/90 min) corresponding to 57.7–66.7% and 57.7–60.8% of total monosaccharides present in the liquid phases from Galega and Cobrançosa pomaces, respectively. For the autohydrolysis conditions at SF = 4.2 (180 °C/120 min) these hydrolysates contained 80.0% and 86.2% oligosaccharides, respectively, with Galega and Cobrançosa pomaces.

In the autohydrolysis of olive tree pruning material carried out by Cara et al., (2012) [38], the obtained OS were mainly XOS and glucooligosaccharides (GlcOS). The highest OS production was between 131–136 g/kg raw material (d.w.) and obtained for temperatures ranging from 170 to 190 °C (10 min isothermal conditions in a Parr reactor). The highest quantities of XOS and GlcOS were obtained at 180 °C (XOS: 60.4 g/kg) and 170 °C (GlcOS: 63.4 g/kg), respectively. Moreover, a sharp decrease in OS production was also observed with increasing temperatures (>190 °C). The OS released, as well as the autohydrolysis optimal temperatures, were similar to those obtained in the present study. In sugarcane bagasse autohydrolysis, performed by Milessi et al. (2021) [39], biomass solubilization increased with the severity factor, reaching a maximum of 59.8% for the highest SF of 5.4 tested (195 °C/20 min). Moreover, xylose release also increased with SF. The lowest xylose concentrations were obtained in pre-treatments at 175 or 185 °C for 10 min (<5%

in the liquid phase). The best autohydrolysis operational condition was 185 °C/10 min (SF = 4.7), resulting in an XOS yield of 45.2% and a xylose yield of 4.4%.

The results presented in Table 5 show that under these hydrothermal conditions, the liquid phases obtained, either from Galega or Cobrançosa pomaces, are very rich in xylooligosaccharides (XOS) with recognized prebiotic properties [6], and contain free xylose which can be valorised to produce xylitol or furfural.

As reported for other materials, e.g., rice straw [40], wheat straw [41], spent brewery grains [42], olive tree pruning material [38], and sugarcane bagasse [39] the hydrolysis of hemicelluloses by hydrothermal processing (autohydrolysis), also results in a high recovery of soluble saccharides in oligomeric forms. However, the optimization of operating conditions is crucial to achieve high oligosaccharide yield.

## 3. Materials and Methods

### 3.1. Raw Materials

Olive pomaces from 'Cobrançosa' (COB) and 'Galega Vulgar' (GAL) olive cultivars (*Olea europaea* L.), were the by-product of olive oil extraction from fruits collected in rainfed organic orchards in the Centre of Portugal, Castelo Branco region (39°50′ N, 7°42′ W). Olives were harvested on three different dates, between 9 October 2020 and 26 November 2020, to obtain olives at different ripening stages. The ripening index (RI) of the olives was assayed according to the guidelines of International Olive Council [43]. The olive oil was immediately extracted from the olives using a laboratory oil-mill (Abencor analyser; MC2 Ingenieria y Sistemas S.L., Seville, Spain). The olives were crushed with a hammer mill equipped with a 4 mm sieve at 3000 rpm. The extraction process was carried out without water addition, simulating the 2-phase decanter. Malaxation of the pastes was performed at 27–30 °C, for 30 min, and centrifugation at 3500 rpm for 1 min. After olive oil extraction, the following samples of olive pomace from 'Cobrançosa' cultivar (COB-1, RI = 2.5; COB-2, RI = 3.3; COB-3, RI = 4.7), and from 'Galega Vulgar' cultivar (GAL-1, RI = 1.8; GAL-2, RI = 2.9; GAL-3, RI = 4.8) were obtained. Pomaces were stored at −18 °C until further use.

### 3.2. Chemical Characterization of the Olive Fruits and Pomaces

#### 3.2.1. Chemical Characterization of the Olive Fruits

Olives from both cultivars with different ripening stages were characterized by the initial moisture and oil content of olive pastes that were assayed by NIR spectroscopy using a spectrometer (MPA, Bruker Optics, Ettlingen, Germany), using the calibration model B-Olive-pastes, Bruker Optics.

#### 3.2.2. Chemical Characterization of Olive Pomaces

Moisture content of pomaces was assayed by oven drying at 100 °C, until constant mass. Prior to chemical analysis, the olive pomaces were dried at 60 °C for 48 h in a laboratory oven. The dried olive pomaces were ground in a cutting mill (Retsch SM 2000; Retsch GmbH, Haan, Germany) to reduce the particle size, using an output sieve of 1.0 mm, and sieved. The 40–60 mesh (0.25–0.42 mm) granulometric fraction was used for chemical analyses.

Ash content was obtained by incinerating the dried pomace samples in a muffle furnace (Heraeus MR 170 E) at 525 °C overnight [44]. The analyses were performed in triplicate, and therefore results were expressed as mean values in percentage of the original dry sample.

The extractives content of dried olive pomaces was determined by sequential extraction in a Soxhlet apparatus with *n*-hexane (6 h), 95% ethanol (16 h) and water (16 h), at a ratio of 1:20 (solid:solvent; *m/v*), using approximately 7 g of dried olive pomace sample in each thimble. The extractives solubilized by each solvent were determined by mass difference of the solid residue after drying at 105 °C and reported as percentage of the original sample. After the first extraction, *n*-hexane was removed, and the oil recovered in

a rotary evaporator. The extracted olive pomaces (EOP) were air dried and stored in a dry place to perform cutin, Klason and acid-soluble lignin analyses.

The cutin content was determined in the extractive-free pomaces by methanolysis for depolymerisation, as described by Simões et al. [45]. A sample of extractive-free material (1.5 g) was refluxed with a 3% (m/v) solution of sodium methoxide in methanol (100 mL) for 3 h. The sample was filtered and washed with methanol, and the residue was refluxed again with 100 mL methanol for 15 min and filtered. After filtration, the combined filtrates were acidified to pH 6 with 2M sulfuric acid and evaporated to dryness. This residue was suspended in water (50 mL) and the alcoholysis products were recovered with dichloromethane in three successive extractions (of 50 mL each). The combined organic extracts were dried over anhydrous sodium sulphate, and the solvent was evaporated to dryness and determined gravimetrically as cutin, which was expressed in percentage of the initial dry mass.

Klason and acid-soluble lignin and carbohydrate contents were determined on the extracted and cutin-free pomaces [46] for Klason lignin and for acid-soluble lignin [47]. Two-step acid hydrolysis was performed subsequently. Three hundred milligrams of dry material were pre-hydrolysed for 60 min with 3 mL of 72% sulfuric acid, which was diluted to a concentration of 3% sulfuric acid. The second step of hydrolysis was performed for 60 min at 120 °C and 0.12 MPa. The sample was filtered through a glass filter crucible (G3) and washed with boiling ultrapure water. Klason lignin was determined as the mass of the solid residue after drying at 105 °C. Acid-soluble lignin was determined on the combined filtrate by measuring the absorbance at 206 nm using a UV–Vis spectrophotometer (Shimadzu UV-160A). Measurements were reported as percentage of the original sample. The total lignin content is the sum of Klason lignin with acid-soluble lignin contents. All analyses were performed in triplicate.

The percentage of polysaccharides was considered as the difference from 100 and the sum of the contents of ash, total extractives, cutin, and total lignin (Klason and acid-soluble).

### 3.3. Autohydrolysis Experiments

The extracted olive pomaces (EOP) were air dried before the hydrothermal pre-treatment, i.e., autohydrolysis. Autohydrolysis was carried out in 100 mL stainless-steel reactors, immersed in an oil bath with controlled temperature and under rotation. The reactors were filled with approximately 3 g of each EOP sample and 60 mL of water, corresponding to liquid-to-solid mass ratio of 20:1 (*v/m*). Autohydrolysis conditions, namely time and temperature, varied according to the experimental design followed (Central Composite Rotatable Design, CCRD) for the optimization of soluble sugar production. Autohydrolysis time varied from 48 to 132 min, and temperature from 142 to 198 °C. A total of 12 runs per EOP (4 factorial points, 4-star points and 4 central points) was performed (Table 6). Factorial points correspond to the experiments 1 to 4, star points correspond to the experiments 5 to 8, and central points correspond to the experiments 9 to 12 [34,35].

At the end of the reaction, the reactors were cooled on ice, to stop the reaction. Afterwards, the liquid phases were filtered using a filter crucible (pore size G3) and collected in a Büchner flask and stored at −18 °C, for further analysis of the sugar content. For each experiment, the severity factor (SF = log $R_0$) was calculated from $R_0$ value, which is given by Equation (1):

$$R_0 = \int_0^t \exp\left(\frac{T - 100}{14.75}\right) dt \tag{1}$$

where the temperature (*T*, °C) is a function of time (*t*, min); 100 °C is the reference temperature and 14.75 °C is a constant regarding the normal energy of activation based on the assumption that, in general, the process is hydrolytic, and the conversion follows first-order kinetics [48]. The experiments of CCRD had a severity factor varying from 3.2 (star point at 142 °C/90 min) to 4.9 (star point at 198 °C/90 min). The concentration of soluble sugars was expressed as glucose equivalent per kg of dry extracted pomace.



**Table 6.** Central Composite Rotatable Design (CCRD) followed for autohydrolysis of extracted olive pomaces with coded and decoded values for the variables (time and temperature).

| Experiment $n^{o}$ | Temperature Coded Value | Time Coded Value | Temperature (°C) | Time (min) |
|---|---|---|---|---|
| 1 | (−1) | (−1) | 150 | 60 |
| 2 | (−1) | 1 | 150 | 120 |
| 3 | 1 | (−1) | 190 | 60 |
| 4 | 1 | 1 | 190 | 120 |
| 5 | $(-\sqrt{2})$ | 0 | 142 | 90 |
| 6 | $\sqrt{2}$ | 0 | 198 | 90 |
| 7 | 0 | $(-\sqrt{2})$ | 170 | 48 |
| 8 | 0 | $\sqrt{2}$ | 170 | 132 |
| 9 | 0 | 0 | 170 | 90 |
| 10 | 0 | 0 | 170 | 90 |
| 11 | 0 | 0 | 170 | 90 |
| 12 | 0 | 0 | 170 | 90 |

*3.4. Sugar Analysis of the Autohydrolysis Liquid Phase*

The quantification of total soluble sugars was carried out using the phenol–sulfuric acid assay [49]. Aliquots of 0.1 mL sample were reacted with 2.5 mL of concentrated sulfuric acid (96%) and 1 mL of phenol solution (5%, *w/v*). The solution was homogenized in a vortex. After cooling, samples were evaluated at absorbances of 490 nm using a spectrophotometer (Shimadzu UV-160A). The calibration curve was obtained with a D-glucose standard solution at six concentrations levels (10–100 mg/L) (six data-points; $R^2$ = 0.987). The results were expressed as g glucose equivalents by kg EOP (d.w.).

For the monosaccharide composition assay, Galega and Cobrançosa liquid samples obtained from the central points (SF = 4.0; 170 °C/90 min) of the CCRD experiments, those from the highest severity factor treatment (SF = 4.9; star point: 198 °C/90 min), and the samples obtained in validation experiments for Galega (GAL-2) and Cobrançosa (COB-2) (SF = 4.4; 180 °C/120 min) were analysed by high-pressure ion-exchange chromatography with a pulsed amperometric detector (HPIC-PAD). The monomeric sugars were separated using a Dionex ICS-3000 system, with an Aminotrap plus Carbopac PA10 column (250 × 4 mm), and a sodium hydroxide/sodium acetate eluent with a 1 mL/min flow at 25 °C. The quantification was performed by external calibration using standard solutions (concentration from 5 ppm to 100 ppm) of the measured compounds (HPLC grade) [19].

The oligosaccharide (OS) content was estimated by the difference between the total sugar content of the liquid phase recovered from the hydrothermal treatment, and the total amount of monosaccharides.

*3.5. Statistical Analysis*

One-way analysis of variance (ANOVA) of the chemical composition results for the pomace samples, as well as the results obtained from autohydrolysis, were performed using the software Statistica, version 7, software from Statsoft, Tulsa, OK, USA. In ANOVA, a post hoc Scheffé test was carried out considering a *p*-value of 0.05.

**4. Conclusions**

This study showed that autohydrolysis, a non-pollutant thermal treatment, can be successfully used for xylo-oligosaccharide (XOS) and xylose monomer production from monovarietal olive pomaces. Response Surface Methodology proved to be a very useful technique to find the most adequate operational conditions (temperature and time) to attain maximum sugar production by autohydrolysis of extracted olive pomaces. Only a set of 12 experiments dictated by the central composite rotatable design, was needed for process optimization, which represents a decrease in experimental costs. The highest soluble sugar production was obtained with high severity treatments of SF ≥ 4.0, corresponding to

temperatures between 175 °C and 200 °C and reaction times between 70 and 130 min, for all pomaces.

Olive pomace hemicelluloses are rich in xylose (>50% of monomeric sugars). Up to 98% yield of XOS was obtained. However, the treatments at SF of 4.9 resulted in a decrease in XOS due to hemicellulose hydrolysis in monomeric units. The results obtained from autohydrolysis were neither dependent on cultivar nor on the ripening stage of the olives. This is an important finding because the results obtained in the present study can be applied to other olive pomaces which can contribute to olive pomace valorisation under the frame of the biorefinery concept.

**Author Contributions:** Conceptualization, S.F.-D., I.M. and F.P.; methodology, S.F.-D., I.M. and F.P.; validation, S.F.-D., I.M. and F.P.; formal analysis, L.F. and R.S.; investigation, L.F. and R.S.; data curation, L.F., R.S., S.F.-D., I.M. and F.P.; writing—original draft preparation, L.F.; writing—review and editing, S.F.-D., I.M. and F.P.; visualization, S.F.-D., I.M. and F.P.; supervision, S.F.-D., I.M. and F.P.; project administration, S.F.-D. All authors have read and agreed to the published version of the manuscript.

**Funding:** This work was funded by FCT—Fundação para a Ciência e a Tecnologia, I.P., through the project LEAF—Linking Landscape, Environment, Agriculture and Food Research Centre (UIDB/04129/2020).

**Data Availability Statement:** Data available upon request.

**Acknowledgments:** The authors thank Conceição Vitorino (IPCB/ESA) for the technical assistance in Abencor extractions. This work was developed under the FCT—Fundação para a Ciência e a Tecnologia, I.P., research units Linking Landscape, Environment, Agriculture and Food Research Centre, LEAF (UIDP/04129/2020; UIDB/04129/2020) and Forest Research Centre (UIDB/00239/2020).

**Conflicts of Interest:** The authors declare no conflict of interest.

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
