# Peer review of "Optimization of Autohydrolysis of Olive Pomaces to Obtain Bioactive Oligosaccharides: The Effect of Cultivar and Fruit Ripening"

_catalysts, doi:10.3390/catal12070788_

Round 1
Reviewer 1 Report
The paper is interesting, well written and the obtained results are satisfactorily statistically analyzed. However, before it can be published, I suggest some important points to be considered:
- Please set clear at line 120 if this moisture characterization is before or after oil extraction
- Table 2 is confusing, what the line “n-hexane” and “ethanol” means? I believe this is an extraction solvent used in some step, however, once in this paper format the methodology comes at the end of manuscript, an introductory paragraph explaining what was done is necessary to make table 2 understandable.
- If the aim of the study was to evaluate the effect of autohydrolysis on hemicellulose solubilization, why authors took glucose and not xylose and xylooligosaccharides as response variables on table 3? This is not clear.
- Line 230 “both T and t”, please write “both temperature and time”
- Line 272 please add units
- Table 5: what is the oligosaccharides concentration in the produced liquor? This is a very important information to compare with other works from literature such the works bellow that I suggest to be included in the discussion to compare olive potential with other biomasses and to compare with other works using olive residues.
https://doi.org/10.1007/s12155-020-10188-7
https://doi.org/10.1016/j.indcrop.2021.114056
https://doi.org/10.1016/j.indcrop.2012.03.017
- Item 3.3 please add the factorial design matrix with the coded variables
- Lines 464-466: please provide references to the methodology used to estimate oligosaccharide %. There are other methods more reliable for that, why authors did not use it?
Author Response
Dear Reviewer,
In my name and of the co-authors, I would like to thank you for the time you spent reviewing our manuscript and your excellent comments that helped us to improve it.
Your suggestions were added to the manuscript and below is the itemized rebuttal to your comments and questions.
Best regards,
Suzana Ferreira-Dias
Comments and Suggestions for Authors
The paper is interesting, well written and the obtained results are satisfactorily statistically analyzed. However, before it can be published, I suggest some important points to be considered:
-Please set clear at line 120 if this moisture characterization is before or after oil extraction
Ans: The moisture characterization is always before oil extraction because after oil extraction we can only determine the water in olive paste. In the manuscript we added “assessed immediately after harvest” to clarify it.
Table 2 is confusing, what the line “n-hexane” and “ethanol” means? I believe this is an extraction solvent used in some step, however, once in this paper format the methodology comes at the end of manuscript, an introductory paragraph explaining what was done is necessary to make table 2 understandable.
Ans: You are right! The presentation of the methodology after results and discussion makes sometimes difficult to understand the results. In fact, the extraction by n-hexane, ethanol and water is a sequential extraction in a Soxhlet apparatus, with the aim of extracting compounds with increasing polarity.
In Table 2, we deleted the line “Total ethanol and water extractives”, that is the sum of the values of the previous lines “n-hexane” and “water”.
As you suggested, the following sentence was added (line 148):
“Total extractives of pomaces were obtained by sequential extraction in a Soxhlet apparatus using solvents with increasing polarity: n-hexane, ethanol and water (c.f. 3.2.2.). The oil content corresponds to n-hexane extracts, while hydrophilic compounds are extracted with ethanol and water.”
If the aim of the study was to evaluate the effect of autohydrolysis on hemicellulose solubilization, why authors took glucose and not xylose and xylooligosaccharides as response variables on table 3? This is not clear.
Ans: the method used to follow sugar release by auto-hydrolysis was the phenol-sulfuric acid assay. It is a spectrophotometric method that is adequate for the quantification of total soluble sugars. The calibration curve was made with glucose and the results were expressed in glucose equivalents (and not glucose). Using xylose for the calibration curve would give similar results. As can be seen in Table 5, xylose is the major sugar but glucose, rhamnose and arabinose are also present. XOS were indirectly estimated from the assay of monomeric sugars by HPLC (c.f. 3). HPLC analysis were carried out in an external laboratory.
-Line 230 “both T and t”, please write “both temperature and time”
Ans: done
Line 272 please add units
Ans: done (g Glu/kg).
Table 5: what is the oligosaccharides concentration in the produced liquor?
Ans: As indicated in Table 5, OS concentration is expressed in mg/g d.w. of extracted olive pomace.
This is a very important information to compare with other works from literature such the works bellow that I suggest to be included in the discussion to compare olive potential with other biomasses and to compare with other works using olive residues.
https://doi.org/10.1007/s12155-020-10188-7
https://doi.org/10.1016/j.indcrop.2021.114056
https://doi.org/10.1016/j.indcrop.2012.03.017
Ans: Thank you very much for this suggestion. We included in the section “2.4 Monomeric and oligosaccharides quantification”, an additional text about the works on XOS production by autohydrolysis of olive tree pruning material (Cara et al., 2012; https://doi.org/10.1016/j.indcrop.2012.03.017) and sugarcane bagasse (Milessi et al., 2021; https://doi.org/10.1016/j.indcrop.2021.114056). Since the other suggested paper (Ávila et al., 2021; https://doi.org/10.1007/s12155-020-10188-7) is about enzymatic production of XOS and not by autohydrolysis, we decided not to include this work in the discussion of our results.
The additional text is as follows:
“In the autohydrolysis of olive tree pruning material carried out by Cara et al., (2012) [38], the obtained OS were mainly XOS and glucooligosaccharides (GlcOS). The highest OS production was between 131-136 g/kg raw material (d.w.) and obtained for temperatures ranging from 170 to 190 °C (10 min isothermal conditions in a Parr reactor). The highest quantities of XOS and GlcOS were obtained at 180 ℃ (XOS: 60.4 g/kg) and 170 ℃ (GlcOS: 63.4 g/kg), respectively. Moreover, a sharp decrease in OS production was also observed with increasing temperatures (>190 ℃). The OS released, as well as the autohydrolysis optimal temperatures, were similar to those obtained in the present study.
In sugarcane bagasse autohydrolysis, performed by Milessi et al. (2021) [39], biomass solubilization increased with the severity factor, reaching a maximum of 59.8 % for the highest SF of 5.4 tested (195 °C/20 min). Moreover, xylose release also increased with SF. The lowest xylose concentrations were obtained in pretreatments at 175 or 185 °C for 10 min (< 5 % in the liquid phase). The best autohydrolysis operational condition was 185 °C/10 min (SF = 4.7), resulting in a XOS yield of 45.2 % and a xylose yield of 4.4 %.”
Item 3.3 please add the factorial design matrix with the coded variables
Ans: The CCRD with coded and decoded values was added as table 6, in section 3.3.
Lines 464-466: please provide references to the methodology used to estimate oligosaccharide %. There are other methods more reliable for that, why authors did not use it?
Ans: The oligosaccharide percentage was estimated as referred in the paper from Cara et al. (2012) you suggested to be cited (https://doi.org/10.1016/j.indcrop.2012.03.017).
In our lab, we don’t have the required methodologies for oligosaccharide analysis. The analyses of monosaccharides were performed by an external laboratory, but that lab could not analyse small oligosaccharides by high-performance size-exclusion chromatography. We tried to search for other laboratories able to make the analysis but those available would take time to implement the technique. Therefore, we decided to use this indirect methodology for OS estimation.
Reviewer 2 Report
The manuscript is well written with adequate references and clearly presented and discussed results. The conclusions are consequently confirmed by the results.
There are some self-citations from the author, but considering they are used to show the earlier experience of the authors in the same topics they are not a big concern. Maybe not all of them are necessary, but it is difficult to decide.
Author Response
Dear Reviewer,
In my name and of the co-authors, I would like to thank you for your comments and the time you spent reviewing our manuscript.
Best regards,
Suzana Ferreira-Dias
Reviewer 3 Report
Please see attached PDF document

Author Response
Dear Reviewer,
In my name and of the co-authors, I would like to thank you for the time you spent reviewing our manuscript and your excellent comments and corrections that helped us to improve it.
Your suggestions were added to the manuscript (track changes) and below is the itemized rebuttal to your comments and questions.
Best regards,
Suzana Ferreira-Dias
Rebuttal:
Introduction
Lines 47-49
Whatever the extraction mill system used, this industry generates large quantities of solid wastes, i.e., the olive pomaces, which contains 3-4.5 % residual oil (wet basis), olive pulp, stones and water.
This does not make sense. What is meant by "what ever the extraction mill system used"
Ans: We want to say that either with pressing or centrifugation olive extraction mills. Thus, we added: (pressing or centrifugation).
Lines 72-77:
Ans: It is plant toxicity.
Line 91-93
“Its purification and valorisation have been an important issue in waste biomass conversion into added-value chemicals [6].”
What does this mean? Please elaborate.
Ans: This sentence was modified as follows:
“Its extraction and purification is needed for waste biomass conversion into added-value chemicals [6].”
Lines 94-95:
However, the use of these different components of lignocellulosic materials needs efficient pre-treatments for biomass fractionation.
Which compounds are referred to here? Please name them.
Ans: The compounds are hemicelluloses, cellulose, and lignin. This information was added.
Lines 104-110
Autohydrolysis has been carried out to produce oligosaccharides and sugars from EOP [19,29,30], or olive stones [19,22]. The extent of hemicellulose hydrolysis depends on the treatment conditions (temperature and time), increasing with the severity of the treatment.
Does not make sense. Please clarify.
Ans: the severity of the treatment is currently evaluated by the parameter “severity factor”, which is defined by eq. 1, in the Material and Methods section. We decided to remove this part of the sentence.
Lines 136-141
Again, only the water content of the pomace obtained from fruits with the highest RI was significantly higher than the values for the other pomaces
Name the cultivar?
Ans: It was observed for both cultivars. The sentence was rewritten as follows:
“Again, for both cultivars, only the water content of the pomaces obtained from fruits with the highest RI was significantly higher than the values for the other pomaces.
Table 2. Chemical composition (% d. w.) of pomaces from Galega (GAL) and Cobrançosa (COB) olives with different ripening indexes (RI). Means, in the same row, followed by the same letters are not significantly different (p > 0.05).
Include the units for the parameters measured.
Ans: As indicated in the heading of Table 2, all the results are expressed in (% d.w.).
Lines 147-157
The chemical composition of pomaces from Galega and Cobrançosa cultivars at different RI is presented in Table 2. The olive pomaces, derived from the Abencor extractor system, are characterized by a high residual oil content (Table ?).
Can't find the "high" residual oil content. How high is high?
Ans: This is a general statement for Abencor extractors yield. The reference [33] was added to this statement.
As said in line 55, industrial olive mills produce “olive pomaces which contain 3-4.5 % residual oil (wet basis), olive pulp, stones, and water.” These values correspond to around 7%- 10 % residual oil (dry basis), considering 55 % moisture content.
In our study, we have 22-31 % residual oil in the pomaces (Table 2: n-hexane extractives), which are, indeed, high values.
Lines 147-157
The residual oil fraction, extracted with n-hexane, represented 25-26 % (d. w.) on Galega pomaces GAL-1 and GAL-2, but greatly increased in GAL-3 (40.56 %). For Cobrançosa pomaces, the value also significantly increased in sample COB-3. Therefore, this residual oil fraction in Galega Vulgar pomaces increased with ripening,…
Ans: In the sentence, “Therefore, this residual oil fraction in Galega Vulgar pomaces increased with ripening”, this increase was not only observed in Galega Vulgar pomace GAL-3 but also in Cobrançosa pomace COB-3. Therefore, we can not accept your suggestion of referring only Galega pomaces. Instead, we wrote “both pomaces”
Lines 158-171
The significant differences found in the concentrations of polar extractives in Galega and Cobrançosa pomaces might be related to with olive cultivar and of the effect of fruit ripening on the formation of these compounds.
Is there a collective name for these compounds? If so, include it.
Ans: they are mainly phenolic compounds. This information was added.
Lines 177-190:
However, the global composition of all samples are within the range of the values reported for extracted olive pomace: Miranda et al. (2019) [19] reported 31.2 % of lignin and 36.5% of polysaccharides (3.8 % of glucan and 22.7 % of hemicelluloses rich in xylans)., Fernandes et al. (2016) [17] reported 33.9 % of Klason lignin, 23.3 % of hemicelluloses and 22.9% of glucan, Leite et al. (2016) [18] reported 41.6 % of lignin and 35.3 % of polysaccharides (24.1 % of hemicelluloses and 11.2% of cellulose), and Ruggeri et al. (2015) [16] reported 37% of lignin, and 49.5 % of polysaccharides in…………...
This does not make sense.
Which olive cultivar?
Ans: These sentences were completed as follows:
“The following values for the composition of extracted olive pomace samples from non-specified cultivars, obtained from industrial olive mills, and partly destoned before pomace oil solvent-extraction, were reported: Miranda et al. (2019) [19] reported 31.2 % of lignin and 36.5% of polysaccharides (3.8 % of glucan and 22.7 % of hemicelluloses rich in xylans), Fernandes et al. (2016) [17] reported 33.9 % of Klason lignin, 23.3 % of hemicelluloses and 22.9% of glucan, Leite et al. (2016) [18] reported 41.6 % of lignin and 35.3 % of polysaccharides (24.1 % of hemicelluloses and 11.2% of cellulose), and Ruggeri et al. (2015) [16] reported 37% of lignin, and 49.5 % of polysaccharides.”
Lines 223-229
GAL-1 and GAL-3 models exhibited a very high fit to the experimental with results since they present determination coefficients higher than 0.90 [34].
Ans: the definition of “very high fit of the model” is presented by Haaland (1989) [34] when the determination coefficient of the model is equal or higher than 0.90. Thus, we cannot remove the word “very”. The determination coefficient is the correlation coefficient to the power of 2. We cannot remove the word “determination” either. These are statistical concepts.
Lines 248-256
Response surfaces presented similar shapes and values for both cultivars. In all situations, the effect of temperature seemed to be more important than that of the time of autohydrolysis.
What does this mean?
Ans: From Table 4 (Polynomial equations of the models fitted to experimental results of autohydrolysis), you can see that for all situations, the coefficients for the linear terms of temperature are always higher than for time. These coefficients do not correspond to the effects of each factor because we are using decoded factors (and not coded factors) in these equations, for easier utilization of the models. Even though, this means that the temperature effect is higher (more important in the response) than the effect of time. Moreover, from the shape of the response surfaces fitted to the experimental data points, we can see that the curvature is more pronounced in relation to the temperature axes and is “almost” parallel to the time axes. For more details, please see the references 34 and 35 or other textbooks on Experimental Design.
The sentence was modified as follows:
“In all situations, the effect of temperature was more important than the effect of time on sugar production. Moreover, the shape of the response surfaces fitted to the experimental data points show that the curvature is more pronounced in relation to the temperature axes than to the time axes.”
Lines 257-267
Analysing the response surfaces and contour plots for the different olive pomaces submitted to autohydrolysis, the best operational regions, in terms of temperature and time, were can be identified (Figures 1 and 2). This is more important, from a technological point of view, than finding the mathematical solution for each model equation, i.e., the stationary point, corresponding to the maximum sugar production.
Rephrase this. What is the "best" operational region?
Ans: For each second-order model equations in Table 4, the optimal conditions of time and temperature to maximize autohydrolysis can be determined by partial differentiation of the polynomial equation and equate the derivative to zero. The solution found for this system of equations is called “stationary point” and corresponds to the maximum (for a convex surface) or to the minimum (for a concave surface) of the response surface. It means that we have a single point as response. However, from an industrial point of view, it is better to know the possible variation range for each factor (time and temperature in this case) that will insure similar responses. This corresponds to the best operation region.
The sentences were modified as follows:
“For each second-order model equations in Table 4, the optimal conditions of time and temperature to maximize sugar production can be determined by partial differentiation of the polynomial equation and equate the derivatives to zero. The mathematical solution found for each system of equations is the stationary point. However, from a technological point of view, it is more important to define the variation range for temperature and time that will lead to high sugar productions, i.e. the best operation region, than to have a single point. Analysing the response surfaces and contour plots for the different olive pomaces submitted to autohydrolysis, the best operational regions, in terms of temperature and time, were identified (Figures 1 and 2)”
Lines 293-302
In our case, where full pomaces were used, higher quantities amounts of olive stones were present. However, our results are in accordance with those previously reported for autohydrolysis of pomaces and stones [References needed].
full pomaces were used, Explain this.
Ans: we used “not destoned pomaces”. “Full pomaces” was replaced by this information.
References were added.
Lines 317-320
The liquid liquors resulting from hydrothermal treatments contained a mixture of oligomeric compounds and monosaccharides. Sugars are predominantly in oligomeric form varying between 80 and 98 % of the total sugars, in Galega liquids liquors, and from 86 to 98 %, in Cobrançosa liquids liquors, depending on the severity conditions of the treatment.
Ans: the term “liquor” is currently used in the field of lignocellulosic technology. However, instead of replacing “liquor” by “liquid”, we prefer to use “liquid phase”.
Lines 331-333
The concentration amount of monomeric xylose reached the highest value at SF = 4.9 (198°C/90 min) corresponding to 57.7-66.7 % and 57.7- 60.8 % of total monosaccharides present in the liquids liquors from Galega and Cobrançosa pomaces, respectively (Table ?).
Ans: All the text in section “2.4 Monomeric and oligosaccharides quantification” is about the results in table 5, as referred in the first line of this section. Thus, we think it is not necessary to repeat “Table 5” in all sentences.
Lines 352-365
The olives were crushed with a hammer mill equipped with a 4 mm sieve at 3000 rpm. Malaxation of the pastes was performed at 27–30 °C, for 30 min, and centrifugation at 3500 rpm for 1 min. The olive oil was extracted from the olives using a laboratory oil-mill (Abencor analyser; MC2 Ingenieria y Sistemas S.L., Seville, Spain). The extraction process was carried out without water addition, simulating the 2-phase decanter. After olive oil extraction, the following samples of olive pomace from ‘Cobrançosa’ cultivar (COB-1, RI=2.5; COB-2, RI = 3.3; COB-3, RI = 4.7), and from ‘Galega Vulgar’ cultivar (GAL-1, RI= 1.8; GAL-2, RI= 2.9; 364 GAL-3, RI = 4.8) were obtained. Pomaces were immediately stored at -18 °C until further use.
This is confusing. Was the olive paste extracted twice? Please clarify.
Ans: The olives were extracted only one time. You are right. The text is confusing. The order of the sentences was changed to clarify the text, as follows:
The olive oil was extracted from the olives using a laboratory oil-mill (Abencor analyser; MC2 Ingenieria y Sistemas S.L., Seville, Spain). The olives were crushed with a hammer mill equipped with a 4 mm sieve at 3000 rpm. The extraction process was carried out without water addition, simulating the 2-phase decanter. Malaxation of the pastes was performed at 27–30 °C, for 30 min, and centrifugation at 3500 rpm for 1 min.
Lines 407-410
mL vs. ml
Ans: mL was used in all the manuscript.
Lines 455-463:
More detail is needed regarding the HPLC technique. Operating temperature, runtime, concentrations of standards for the calibration curve etc. A reference is also needed for the HPLC technique used.
Ans: The following information was added:
“The monomeric sugars were separated using a Dionex ICS-3000 system, with an Ami-notrap plus Carbopac PA10 column (250 x 4 mm), and a sodium hydroxide/sodium acetate eluent with a 1 mL/min flow at 25 °C. The quantification was performed by external calibration using standard solutions (concentration from 5 ppm to 100 ppm) of the measured compounds (HPLC grade) [19].”
Conclusions:
Process optimization by response surface methodology was of utmost importance to establish find the optimum operational conditions (temperature and time) to attain maximum sugar production.
These words are meaningless. Say what the role was of optimization by response surface methodology.
Ans: The sentence was replaced by the following sentence:
“Response Surface methodology was a very useful technique to find the most adequate operational conditions (temperature and time) to attain maximum sugar production by autohydrolysis of extracted olive pomaces. Only a set of 12 experiments dictated by the central composite rotatable design, was needed for process optimization, which represents a decrease in experimental costs.”
Round 2
Reviewer 1 Report
The authors fulfilled all requests correctly